# Restricted Boltzmann machines modeling human choice

**Takayuki Osogami**
IBM Research - Tokyo
osogami@jp.ibm.com

**Makoto Otsuka**
IBM Research - Tokyo
motsuka@ucla.edu

## Abstract

We extend the multinomial logit model to represent some of the empirical phenomena that are frequently observed in the choices made by humans. These phenomena include the similarity effect, the attraction effect, and the compromise effect. We formally quantify the strength of these phenomena that can be represented by our choice model, which illuminates the flexibility of our choice model. We then show that our choice model can be represented as a restricted Boltzmann machine and that its parameters can be learned effectively from data. Our numerical experiments with real data of human choices suggest that we can train our choice model in such a way that it represents the typical phenomena of choice.

## 1 Introduction

Choice is a fundamental behavior of humans and has been studied extensively in Artificial Intelligence and related areas. The prior work suggests that the choices made by humans can significantly depend on available alternatives, or the choice set, in rather complex but systematic ways [13]. The empirical phenomena that result from such dependency on the choice set include the similarity effect, the attraction effect, and the compromise effect. Informally, the similarity effect refers to the phenomenon that a new product, $S$, reduces the share of a similar product, $A$, more than a dissimilar product, $B$ (see Figure 1 (a)). With the attraction effect, a new dominated product, $D$, increases the share of the dominant product, $A$ (see Figure 1 (b)). With the compromise effect, a product, $C$, has a relatively larger share when two extreme products, $A$ and $B$, are in the market than when only one of $A$ and $B$ is in the market (see Figure 1 (c)). We call these three empirical phenomena as the *typical choice phenomena*.

However, the standard choice model of the multinomial logit model (MLM) and its variants cannot represent at least one of the typical choice phenomena [13]. More descriptive models have been proposed to represent the typical choice phenomena in some representative cases [14, 19]. However, it is unclear when and to what degree the typical choice phenomena can be represented. Also, no algorithms have been proposed for training these descriptive models from data.

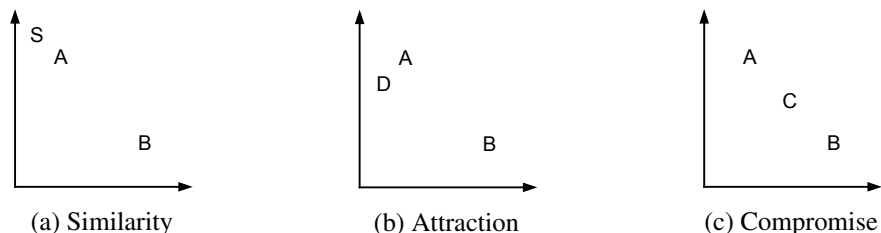

(a) Similarity     (b) Attraction     (c) Compromise

Figure 1: Choice sets that cause typical choice phenomena.

We extend the MLM to represent the typical choice phenomena, which is our first contribution. We show that our choice model can be represented as a restricted Boltzmann machine (RBM). Our choice model is thus called the RBM choice model. An advantage of this representation as an RBM is that training algorithms for RBMs are readily available. See Section 2.

We then formally define the measure of the strength for each typical choice phenomenon and quantify the strength of each typical choice phenomenon that the RBM choice model can represent. Our analysis not only gives a guarantee on the flexibility of the RBM choice model but also illuminates why the RBM choice model can represent the typical choice phenomena. These definitions and analysis constitute our second contribution and are presented in Section 3.

Our experiments suggest that we can train the RBM choice model in such a way that it represents the typical choice phenomena. We show that the trained RBM choice model can then adequately predict real human choice on the means of transportation [2]. These experimental results constitute our third contribution and are presented in Section 4.

## 2 Choice model with restricted Boltzmann machine

We extend the MLM to represent the typical choice phenomena. Let $\mathcal{I}$ be the set of items. For $A \in \mathcal{X} \subseteq \mathcal{I}$, we study the probability that an item, $A$, is selected from a choice set, $\mathcal{X}$. This probability is called the choice probability. The model of choice, equipped with the choice probability, is called a choice model. We use $A, B, C, D, S$, or $X$ to denote an item and $\mathcal{X}, \mathcal{Y}$, or a set such as $\{A, B\}$ to denote a choice set.

For the MLM, the choice probability of $A$ from $\mathcal{X}$ can be represented by

$$p(A|\mathcal{X}) \;\; = \;\; \frac{\lambda(A|\mathcal{X})}{\sum_{X \in \mathcal{X}} \lambda(X|\mathcal{X})}, \tag{1}$$

where we refer to $\lambda(X|\mathcal{X})$ as the choice rate of $X$ from $\mathcal{X}$. The choice rate of the MLM is given by

$$\lambda^{\mathrm{MLM}}(X|\mathcal{X}) \;\; = \;\; \exp(b_X), \tag{2}$$

where $b_X$ can be interpreted as the attractiveness of $X$. One could define $b_X$ through $u_X$, the vector of the utilities of the attributes for $X$, and $\alpha$, the vector of the weight on each attribute (i.e., $b_X \equiv \alpha \cdot u_X$). Observe that $\lambda^{\mathrm{MLM}}(X|\mathcal{X})$ is independent of $\mathcal{X}$ as long as $X \in \mathcal{X}$. This independence causes the incapability of the MLM in representing the typical choice phenomena.

We extend the choice rate of (2) but keep the choice probability in the form of (1). Specifically, we consider the following choice rate:

$$\lambda(X|\mathcal{X}) \equiv \exp(b_X) \prod_{k \in \mathcal{K}} \left( 1 + \exp\left( T_{\mathcal{X}}^k + U_X^k \right) \right), \tag{3}$$

where we define

$$T_{\mathcal{X}}^k \equiv \sum_{Y \in \mathcal{X}} T_Y^k. \tag{4}$$

Our choice model has parameters, $b_X, T_{\mathcal{X}}^k, U_X^k$ for $X \in \mathcal{X}, k \in \mathcal{K}$, that take values in $(-\infty, \infty)$. Equation (3) modifies $\exp(b_X)$ by multiplying factors. Each factor is associated with an index, $k$, and has parameters, $T_{\mathcal{X}}^k$ and $U_X^k$, that depend on $k$. The set of these indices is denoted by $\mathcal{K}$.

We now show that our choice model can be represented as a restricted Boltzmann machine (RBM). This means that we can use existing algorithms for RBMs to learn the parameters of the RBM choice model (see Appendix A.1).

An RBM consists of a layer of visible units, $i \in \mathcal{V}$, and a layer of hidden units, $k \in \mathcal{H}$. A visible unit, $i$, and a hidden unit, $k$, are connected with weight, $W_i^k$. The units within each layer are disconnected from each other. Each unit is associated with a bias. The bias of a visible unit, $i$, is denoted by $b_i^{\mathrm{vis}}$. The bias of a hidden unit, $k$, is denoted by $b_k^{\mathrm{hid}}$. A visible unit, $i$, is associated with a binary variable, $z_i$, and a hidden unit, $k$, is associated with a binary variable, $h_k$, which takes a value in $\{0, 1\}$.

For a given configuration of binary variables, the energy of the RBM is defined as

$$E_\theta(z, h) \equiv -\sum_{i \in \mathcal{V}} \sum_{k \in \mathcal{H}} \left( z_i \, W_i^k \, h_k + b_i^{\mathrm{vis}} \, z_i + b_k^{\mathrm{hid}} \, h_k \right), \tag{5}$$

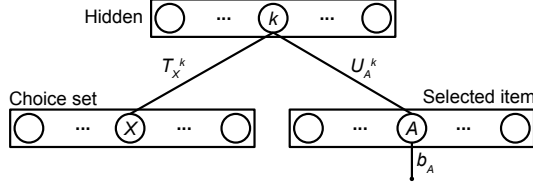

Figure 2: RBM choice model

where $\theta \equiv \{W, b^{\mathrm{vis}}, b^{\mathrm{hid}}\}$ denotes the parameters of the RBM. The probability of realizing a particular configuration of $(z, h)$ is given by

$$P_\theta(z, h) \equiv \frac{\exp(-E_\theta(z, h))}{\sum_{z'} \sum_{h'} \exp(-E_\theta(z', h'))}. \tag{6}$$

The summation with respect to a binary vector (i.e., $\sum_{z'}$ or $\sum_{h'}$) denotes the summation over all of the possible binary vectors of a given length. The length of $z'$ is $|\mathcal{V}|$, and the length of $h'$ is $|\mathcal{H}|$.

The RBM choice model can be represented as an RBM having the structure in Figure 2. Here, the layer of visible units is split into two parts: one for the choice set and the other for the selected item. The corresponding binary vector is denoted by $z = (v, w)$. Here, $v$ is a binary vector associated with the part for the choice set. Specifically, $v$ has length $|\mathcal{I}|$, and $v_X = 1$ denotes that $X$ is in the choice set. Analogously, $w$ has length $|\mathcal{I}|$, and $w_A = 1$ denotes that $A$ is selected. We use $T_X^k$ to denote the weight between a hidden unit, $k$, and a visible unit, $X$, for the choice set. We use $U_A^k$ to denote the weight between a hidden unit, $k$, and a visible unit, $A$, for the selected item. The bias is zero for all of the hidden units and for all of the visible units for the choice set. The bias for a visible unit, $A$, for the selected item is denoted by $b_A$. Finally, let $\mathcal{H} = \mathcal{K}$.

The choice rate (3) of the RBM choice model can then be represented by

$$\lambda(A|\mathcal{X}) = \sum_h \exp\left(-E_\theta\left(\left(v^{\mathcal{X}}, w^A\right), h\right)\right), \tag{7}$$

where we define the binary vectors, $v^{\mathcal{X}}, w^A$, such that $v_i^{\mathcal{X}} = 1$ iff $i \in \mathcal{X}$ and $w_j^A = 1$ iff $j = A$. Observe that the right-hand side of (7) is

$$
\begin{aligned}
\sum_h \exp(-E_\theta((v^{\mathcal{X}}, w^A), h)) &= \sum_h \exp\left(\sum_{X \in \mathcal{X}} \sum_k T_X^k h_k + \sum_k U_A^k h_k + b_A\right) & (8) \\
&= \exp(b_A) \sum_h \prod_k \exp\left(\left(T_{\mathcal{X}}^k + U_A^k\right) h_k\right) & (9) \\
&= \exp(b_A) \prod_k \sum_{h_k \in \{0,1\}} \exp\left(\left(T_{\mathcal{X}}^k + U_A^k\right) h_k\right), & (10)
\end{aligned}
$$

which is equivalent to (3).

The RBM choice model assumes that one item from a choice set is selected. In the context of the RBM, this means that $w_A = 1$ for only one $A \in \mathcal{X} \subseteq \mathcal{I}$. Using (6), our choice probability (1) can be represented by

$$p(A|\mathcal{X}) = \frac{\sum_h P_\theta((v^{\mathcal{X}}, w^A), h)}{\sum_{X \in \mathcal{X}} \sum_h P_\theta((v^{\mathcal{X}}, w^X), h)}. \tag{11}$$

This is the conditional probability of realizing the configuration, $(v^{\mathcal{X}}, w^A)$, given that the realized configuration is either of the $(v^{\mathcal{X}}, w^X)$ for $X \in \mathcal{X}$. See Appendix A.2 for an extension of the RBM choice model.

## 3 Flexibility of the RBM choice model

In this section, we formally study the flexibility of the RBM choice model. Recall that $\lambda(X|\mathcal{X})$ in (3) is modified from $\lambda^{\mathrm{MLM}}(X|\mathcal{X})$ in (2) by a factor,

$$1 + \exp\left(T_{\mathcal{X}}^k + U_X^k\right), \tag{12}$$

for each $k$ in $\mathcal{K}$, so that $\lambda(X|\mathcal{X})$ can depend on $\mathcal{X}$ through $T_{\mathcal{X}}^k$. We will see how this modification allows the RBM choice model to represent each of the typical choice phenomena.

The similarity effect refers to the following phenomenon [14]:

$$p(A|\{A,B\}) > p(B|\{A,B\}) \quad \text{and} \quad p(A|\{A,B,S\}) < p(B|\{A,B,S\}). \tag{13}$$

Motivated by (13), we define the strength of the similarity effect as follows:

**Definition 1.** *For $A, B \in \mathcal{X}$, the strength of the similarity effect of $S$ on $A$ relative to $B$ with $\mathcal{X}$ is defined as follows:*

$$\psi_{A,B,S,\mathcal{X}}^{(\text{sim})} \equiv \frac{p(A|\mathcal{X})}{p(B|\mathcal{X})} \frac{p(B|\mathcal{X} \cup \{S\})}{p(A|\mathcal{X} \cup \{S\})}. \tag{14}$$

When $\psi_{A,B,S,\mathcal{X}}^{(\text{sim})} = 1$, adding $S$ into $\mathcal{X}$ does not change the ratio between $p(A|\mathcal{X})$ and $p(B|\mathcal{X})$. Namely, there is no similarity effect. When $\psi_{A,B,S,\mathcal{X}}^{(\text{sim})} > 1$, we can increase $\frac{p(B|\mathcal{X})}{p(A|\mathcal{X})}$ by a factor of $\psi_{A,B,S,\mathcal{X}}^{(\text{sim})}$ by the addition of $S$ into $\mathcal{X}$. This corresponds to the similarity effect of (13). When $\psi_{A,B,S,\mathcal{X}}^{(\text{sim})} < 1$, this ratio decreases by an analogous factor. We will study the strength of this (rather general) similarity effect without the restriction that $S$ is "similar" to $A$ (see Figure 1 (a)).

Because $p(X|\mathcal{X})$ has a common denominator for $X = A$ and $X = B$, we have

$$\psi_{A,B,S,\mathcal{X}}^{(\text{sim})} = \frac{\lambda(A|\mathcal{X})}{\lambda(B|\mathcal{X})} \frac{\lambda(B|\mathcal{X} \cup \{S\})}{\lambda(A|\mathcal{X} \cup \{S\})}. \tag{15}$$

The MLM cannot represent the similarity effect, because the $\lambda^{\text{MLM}}(X|\mathcal{X})$ in (2) is independent of $\mathcal{X}$. For any choice sets, $\mathcal{X}$ and $\mathcal{Y}$, we must have

$$\frac{\lambda^{\text{MLM}}(A|\mathcal{X})}{\lambda^{\text{MLM}}(B|\mathcal{X})} = \frac{\lambda^{\text{MLM}}(A|\mathcal{Y})}{\lambda^{\text{MLM}}(B|\mathcal{Y})}. \tag{16}$$

The equality (16) is known as the *independence from irrelevant alternatives* (IIA).

The RBM choice model can represent an arbitrary strength of the similarity effect. Specifically, by adding an element, $\hat{k}$, into $\mathcal{K}$ of (3), we can set $\frac{\lambda(A|\mathcal{X} \cup \{S\})}{\lambda(A|\mathcal{X})}$ at an arbitrary value without affecting the value of $\lambda(B|\mathcal{Y}), \forall B \neq A$, for any $\mathcal{Y}$. We prove the following theorem in Appendix C:

**Theorem 1.** *Consider an RBM choice model where the choice rate of $X$ from $\mathcal{X}$ is given by (2). Let $\hat{\lambda}(X|\mathcal{X})$ be the corresponding choice rate after adding $\hat{k}$ into $\mathcal{K}$. Namely,*

$$\hat{\lambda}(X|\mathcal{X}) = \lambda(X|\mathcal{X}) \left(1 + \exp\left(T_{\mathcal{X}}^{\hat{k}} + U_X^{\hat{k}}\right)\right). \tag{17}$$

*Consider an item $A \in \mathcal{X}$ and an item $S \notin \mathcal{X}$. For any $c \in (0, \infty)$ and $\varepsilon > 0$, we can then choose $T_{\cdot}^{\hat{k}}$ and $U_{\cdot}^{\hat{k}}$ such that*

$$c = \frac{\hat{\lambda}(A|\mathcal{X} \cup \{S\})}{\hat{\lambda}(A|\mathcal{X})}; \qquad \varepsilon > \left|\frac{\hat{\lambda}(B|\mathcal{Y})}{\lambda(B|\mathcal{Y})} - 1\right|, \ \forall \mathcal{Y}, B \text{ s.t. } B \neq A. \tag{18}$$

By (15) and Theorem 1, the strength of the similarity effect after adding $\hat{k}$ into $\mathcal{K}$ is

$$\hat{\psi}_{A,B,S,\mathcal{X}}^{(\text{sim})} = \frac{\hat{\lambda}(A|\mathcal{X})}{\hat{\lambda}(A|\mathcal{X} \cup \{S\})} \frac{\hat{\lambda}(B|\mathcal{X} \cup \{S\})}{\hat{\lambda}(B|\mathcal{X})} \approx \frac{1}{c} \frac{\lambda(B|\mathcal{X} \cup \{S\})}{\lambda(B|\mathcal{X})}. \tag{19}$$

Because $c$ can take an arbitrary value in $(0, \infty)$, the additional factor, (12) with $k = \hat{k}$, indeed allows $\hat{\psi}_{A,B,S,\mathcal{X}}^{(\text{sim})}$ to take any positive value without affecting the value of $\lambda(B|\mathcal{Y}), \forall B \neq A$, for any $\mathcal{Y}$. The first part of (18) guarantees that this additional factor does not change $p(X|\mathcal{Y})$ for any $X$ if $A \notin \mathcal{Y}$. Note that what we have shown is not limited to the similarity effect of (13). The RBM choice model can represent an arbitrary phenomenon where the choice set affects the ratio of the choice rate.

According to [14], the attraction effect is represented by

$$p(A|\{A, B\}) < p(A|\{A, B, D\}). \tag{20}$$

The MLM cannot represent the attraction effect, because the $\lambda^{\mathrm{MLM}}(X|\mathcal{Y})$ in (2) is independent of $\mathcal{Y}$, and we must have $\sum_{X \in \mathcal{X}} \lambda^{\mathrm{MLM}}(X|\mathcal{X}) \leq \sum_{X \in \mathcal{Y}} \lambda^{\mathrm{MLM}}(X|\mathcal{Y})$ for $\mathcal{X} \subset \mathcal{Y}$, which in turn implies the *regularity principle*: $p(X|\mathcal{X}) \geq p(X|\mathcal{Y})$ for $\mathcal{X} \subset \mathcal{Y}$.

Motivated by (20), we define the strength of the attraction effect as the magnitude of the change in the choice probability of an item when another item is added into the choice set. Formally,

**Definition 2.** *For $A \in \mathcal{X}$, the strength of the attraction effect of $D$ on $A$ with $\mathcal{X}$ is defined as follows:*

$$\psi_{A,D,\mathcal{X}}^{(\mathrm{att})} \equiv \frac{p(A|\mathcal{X} \cup \{D\})}{p(A|\mathcal{X})}. \tag{21}$$

When there is no attraction effect, adding $D$ into $\mathcal{X}$ can only decrease $p(A|\mathcal{X})$; hence, $\psi_{A,D,\mathcal{X}}^{(\mathrm{att})} \leq 1$. The standard definition of the attraction effect (20) implies $\psi_{A,D,\mathcal{X}}^{(\mathrm{att})} > 1$. We study the strength of this attraction effect without the restriction that $A$ "dominates" $D$ (see Figure 1 (b)).

We prove the following theorem in Appendix C:

**Theorem 2.** *Consider the two RBM choice models in Theorem 1. The first RBM choice model has the choice rate given by (3), and the second RBM choice model has the choice rate given by (17). Let $p(\cdot|\cdot)$ denote the choice probability for the first RBM choice model and $\hat{p}(\cdot|\cdot)$ denote the choice probability for the second RBM choice model. Consider an item $A \in \mathcal{X}$ and an item $D \notin \mathcal{X}$. For any $r \in (p(A|\mathcal{X} \cup \{D\}), 1/p(A|\mathcal{X}))$ and $\varepsilon > 0$, we can choose $T^{\hat{k}}, U^{\hat{k}}$ such that*

$$r = \frac{\hat{p}(A|\mathcal{X} \cup \{D\})}{\hat{p}(A|\mathcal{X})}; \qquad \varepsilon > \left| \frac{\hat{\lambda}(B|\mathcal{Y})}{\lambda(B|\mathcal{Y})} - 1 \right|, \ \forall \mathcal{Y}, B \ \text{s.t.} \ B \neq A. \tag{22}$$

We expect that the range, $(p(A|\mathcal{X} \cup \{D\}), 1/p(A|\mathcal{X}))$, of $r$ in the theorem covers the attraction effect in practice. Also, this range is the widest possible in the following sense. The factor (12) can only increase $\lambda(X|\mathcal{Y})$ for any $X, \mathcal{Y}$. The form of (1) then implies that, to decrease $p(A|\mathcal{Y})$, we must increase $\lambda(X|\mathcal{Y})$ for $X \neq A$. However, increasing $\lambda(X|\mathcal{Y})$ for $X \neq A$ is not allowed due to the second part of (22) with $\varepsilon \to 0$. Namely, the additional factor, (12) with $k = \hat{k}$, can only increase $p(A|\mathcal{Y})$ for any $\mathcal{Y}$ under the condition of the second part of (22). The lower limit, $p(A|\mathcal{X} \cup \{D\})$, is achieved when $\hat{p}(A|\mathcal{X}) \to 1$, while keeping $\hat{p}(A|\mathcal{X} \cup \{D\}) \approx p(A|\mathcal{X} \cup \{D\})$. The upper limit, $1/p(A|\mathcal{X})$, is achieved when $\hat{p}(A|\mathcal{X} \cup \{D\}) \to 1$, while keeping $\hat{p}(A|\mathcal{X}) \approx p(A|\mathcal{X})$.

According to [18], the compromise effect is formally represented by

$$\frac{p(C|\{A, B, C\})}{\displaystyle\sum_{X \in \{A,C\}} p(X|\{A, B, C\})} > p(C|\{A, C\}) \ \text{and} \ \frac{p(C|\{A, B, C\})}{\displaystyle\sum_{X \in \{B,C\}} p(X|\{A, B, C\})} > p(C|\{B, C\}). \tag{23}$$

The MLM cannot represent the compromise effect, because the $\lambda^{\mathrm{MLM}}(X|\mathcal{Y})$ in (2) is independent of $\mathcal{Y}$, which in turn makes the inequalities in (23) equalities.

Motivated by (23), we define the strength of the compromise effect as the magnitude of the change in the conditional probability of selecting an item, $C$, given that either $C$ or another item, $A$, is selected when yet another item, $B$, is added into the choice set. More precisely, we also exchange the roles of $A$ and $B$, and study the minimum magnitude of those changes:

**Definition 3.** *For a choice set, $\mathcal{X}$, and items, $A, B, C$, such that $A, B, C \in \mathcal{X}$, let*

$$\phi_{A,B,C,\mathcal{X}} \equiv \frac{q_{AC}(C|\mathcal{X})}{q_{AC}(C|\mathcal{X} \setminus \{B\})}, \tag{24}$$

*where, for $\mathcal{Y}$ such that $A, C \in \mathcal{Y}$, we define*

$$q_{AC}(C|\mathcal{Y}) \equiv \frac{p(C|\mathcal{Y})}{\sum_{X \in \{A,C\}} p(X|\mathcal{Y})}. \tag{25}$$

*The strength of the compromise effect of $A$ and $B$ on $C$ with $\mathcal{X}$ is then defined as*

$$\psi_{A,B,C,\mathcal{X}}^{(\mathrm{com})} \equiv \min \{\phi_{A,B,C,\mathcal{X}}, \phi_{B,A,C,\mathcal{X}}\}. \tag{26}$$

Here, we do not have the restriction that $C$ is a "compromise" between $A$ and $B$ (see Figure 1 (c)).

In Appendix C:we prove the following theorem:

**Theorem 3.** *Consider a choice set, $\mathcal{X}$, and three items, $A, B, C \in \mathcal{X}$. Consider the two RBM choice models in Theorem 2. Let $\hat{\psi}_{A,B,C,\mathcal{X}}^{(\text{com})}$ be defined analogously to (26) but with $\hat{p}(\cdot|\cdot)$. Let*

$$\overline{q} \equiv \max\left\{q_{AC}(C|\mathcal{X} \setminus \{B\}), q_{BC}(C|\mathcal{X} \setminus \{A\})\right\} \tag{27}$$

$$\underline{q} \equiv \min\left\{q_{AC}(C|\mathcal{X}), q_{BC}(C|\mathcal{X})\right\}. \tag{28}$$

*Then, for any $r \in (\underline{q}, 1/\overline{q})$ and $\varepsilon > 0$, we can choose $T^k, U^k$ such that*

$$r = \hat{\psi}_{A,B,C,\mathcal{X}}^{(\text{com})}; \qquad \varepsilon > \left|\frac{\hat{\lambda}(X|\mathcal{Y})}{\lambda(X|\mathcal{Y})} - 1\right|, \ \forall \mathcal{Y}, X \text{ s.t. } X \neq C. \tag{29}$$

We expect that the range of $r$ in the theorem covers the compromising effect in practice. Also, this range is best possible in the sense analogous to what we have discussed with the range in Theorem 2. Because the additional factor, (12) with $k = \hat{k}$, can only increase $p(C|\mathcal{Y})$ for any $\mathcal{Y}$ under the condition of the second part of (29), it can only increase $q_{XC}(C|\mathcal{Y})$ for $X \in \{A, B\}$. The lower limit, $\underline{q}$, is achieved when $q_{XC}(C|\mathcal{X} \setminus \{X\}) \to 1$, while keeping $q_{XC}(C|\mathcal{X})$ approximately unchanged, for $X \in \{A, B\}$. The upper limit, $1/\overline{q}$, is achieved when $q_{XC}(C|\mathcal{X}) \to 1$, while keeping $q_{XC}(C|\mathcal{X} \setminus \{X\})$ approximately unchanged, for $X \in \{A, B\}$.

## 4 Numerical experiments

We now validate the effectiveness of the RBM choice model in predicting the choices made by humans. Here we use the dataset from [2], which is based on the survey conducted in Switzerland, where people are asked to choose a means of transportation from given options. A subset of the dataset is used to train the RBM choice model, which is then used to predict the choice in the remaining dataset. In Appendix B.2,we also conduct an experiment with artificial dataset and show that the RBM choice model can indeed be trained to represent each of the typical choice phenomena. This flexibility in the representation is the basis of the predictive accuracy of the RBM choice model to be presented in this section. All of our experiments are run on a single core of a Windows PC with main memory of 8 GB and Core i5 CPU of 2.6 GHz.

The dataset [2] consists of 10,728 choices that 1,192 people have made from a varying choice set. For those who own a car, the choice set has three items: a train, a maglev, and a car. For those who do not own a car, the choice set consists of a train and a maglev. The train can operate at the interval of 30, 60, or 120 minutes. The maglev can operate at the interval of 10, 20, or 30 minutes. The trains (or maglevs) with different intervals are considered to be distinct items in our experiment.

Figure 3 (a) shows the empirical choice probability for each choice set. Each choice set consists of a train with a particular interval (blue, shaded) and a maglev with a particular interval (red, mesh) possibly with a car (yellow, circles). The interval of the maglev varies as is indicated at the bottom of the figure. The interval of the train is indicated at the left side of the figure. For each combination of the intervals of the train and the maglev, there are two choice sets, with or without a car.

We evaluate the accuracy of the RBM choice model in predicting the choice probability for an arbitrary choice set, when the RBM choice model is trained with the data of the choice for the remaining 17 choice sets (i.e., we have 18 test cases). We train the RBM choice model (or the MLM) by the use of discriminative training with stochastic gradient descent using the mini-batch of size 50 and the learning rate of $\eta = 0.1$ (see Appendix A.1).Each run of the evaluation uses the entire training dataset 50 times for training, and the evaluation is repeated five times by varying the initial values of the parameters. The elements of $T$ and $U$ are initialized independently with samples from the uniform distribution on $[-10/\sqrt{\max(|\mathcal{I}|, |\mathcal{K}|)}, -10/\sqrt{\max(|\mathcal{I}|, |\mathcal{K}|)}]$, where $|\mathcal{I}| = 7$ is the number of items under consideration, and $|\mathcal{K}|$ is the number of hidden nodes. The elements of $b$ are initialized with samples from the uniform distribution on $[-1, 1]$.

Figure 3 (b) shows the Kullback-Leibler (KL) divergence between the predicted distribution of the choice and the corresponding true distribution. The dots connected with a solid line show the the

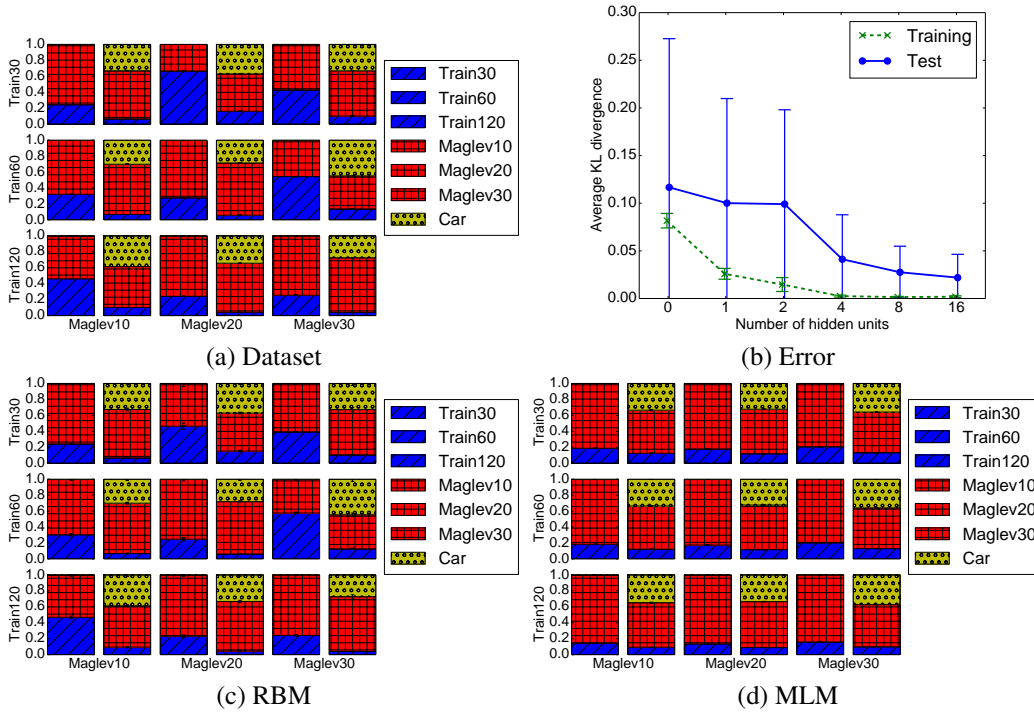

(a) Dataset

(b) Error

(c) RBM

(d) MLM

Figure 3: Dataset (a), the predictive error of the RBM choice model against the number of hidden units (b), and the choice probabilities learned by the RBM choice model (c) and the MLM (d).

average KL divergence over all of the 18 test cases and five runs with varying initialization. The average KL divergence is also evaluated for training data and is shown with a dashed line. The confidence interval represents the corresponding standard deviation. The wide confidence interval is largely due to the variance between test instances (see Figure 4 in the appendix. The horizontal axis shows the number of the hidden units in the RBM choice model, where zero hidden units correspond to the MLM. The average KL divergence is reduced from 0.12 for the MLM to 0.02 for the RBM choice model with 16 hidden units, an improvement by a factor of six.

Figure 3 (c)-(d) shows the choice probabilities given by (a) the RBM choice model with 16 hidden units and (b) the MLM, after these models are trained for the test case where the choice set consists of the train with 30-minute interval (Train30) and the maglev with 20-minute interval (Maglev20). Observe that the RBM choice model gives the choice probabilities that are close to the true choice probabilities shown in Figure 3 (a), while the MLM has difficulty in fitting these choice probabilities. Taking a closer look at Figure 3 (a), we can observe that the MLM is fundamentally incapable of learning this dataset. For example, Train30 is more popular than Maglev20 for people who do not own cars, while the preference is reversed for car owners (i.e., the attraction effect). The attraction effect can also be seen for the combination of Maglev30 and Train60. As we have discussed in Section 3, the MLM cannot represent such attraction effects, but the RBM choice model can.

## 5   Related work

We now review the prior work related to our contributions. We will see that all of the existing choice models either cannot represent at least one of the typical choice phenomena or do not have systematic training algorithms. We will also see that the prior work has analyzed choice models with respect to whether those choice models can represent typical choice phenomena or others but only in specific cases of specific strength. On the contrary, our analysis shows that the RBM choice model can represent the typical choice phenomena for all cases of the specified strength.

A majority of the prior work on the choice model is about the MLM and its variants such as the hierarchical MLM [5], the multinomial probit model [6], and, generally, random utility models [17].

In particular, the attraction effect cannot be represented by these variants of the MLM [13]. In general, when the choice probability depends only on the values that are determined independently for each item (e.g., the models of [3, 7]), none of the typical choice phenomena can be represented [18]. Recently, Hruschka has proposed a choice model based on an RBM [9], but his choice model cannot represent any of the typical choice phenomena, because the corresponding choice rate is independent of the choice set. It is thus nontrivial how we use the RBM as a choice model in such a way that the typical choice phenomena can be represented. In [11], a hierarchical Bayesian choice model is shown to represent the attraction effect in a specific case.

There also exist choice models that have been numerically shown to represent all of the typical choice phenomena for some specific cases. For example, sequential sampling models, including the decision field theory [4] and the leaky competing accumulator model [19], are meant to directly mimic the cognitive process of the human making a choice [12]. However, no paper has shown an algorithm that can train a sequential sampling model in such a way that the trained model exhibits the typical choice phenomena. Shenoy and Yu propose a hierarchical Bayesian model to represent the three typical choice phenomena [16]. Although they perform inferences of the posterior distributions that are needed to compute the choice probabilities with their model, they do not show how to train their model to fit the choice probabilities to given data. Their experiments show that their model represents the typical choice phenomena in particular cases, where the parameters of the model are set manually. Rieskamp et al. classify choice models according to whether a choice model can never represent a certain phenomenon or can do so in some cases to some degree [13]. The phenomena studied in [13] are not limited to the typical choice phenomena, but they list the typical choice phenomena as the ones that are robust and significant. Also, Otter et al. exclusively study all of the typical choice phenomena [12].

Luce is a pioneer of the formal analysis of choice models, which however is largely qualitative [10]. For example, Lemma 3 of [10] can tell us whether a given choice model satisfies the IIA in (16) for all cases or it violates the IIA for some cases to some degree. We address the new question of to what degree a choice model can represent each of the typical choice phenomena (e.g., to what degree the RBM choice model can violate the IIA).

Finally, our theorems can be contrasted with the universal approximation theorem of RBMs, which states that an arbitrary distribution can be approximated arbitrarily closely with a sufficient number of hidden units [15, 8]. This is in contrast to our theorems, which show that a single hidden unit suffices to represent the typical choice phenomena of the strength that is specified in the theorems.

## 6   Conclusion

The RBM choice model is developed to represent the typical choice phenomena that have been reported frequently in the literature of cognitive psychology and related areas. Our work motivates a new direction of research on using RBMs to model such complex behavior of humans. Particularly interesting behavior includes the one that is considered to be irrational or the one that results from cognitive biases (see e.g. [1]). The advantages of the RBM choice model that are demonstrated in this paper include their flexibility in representing complex behavior and the availability of effective training algorithms.

The RBM choice model can incorporate the attributes of the items in its parameters. Specifically, one can represent the parameters of the RBM choice model as functions of $u_X$, the attributes of $X \in \mathcal{I}$ analogously to the MLM, where $b_X$ can be represented as $b_X = \alpha \cdot u_X$ as we have discussed after (2). The focus of this paper is in designing the fundamental structure of the RBM choice model and analyzing its fundamental properties, and the study about the RBM choice model with attributes will be reported elsewhere. Although the attributes are important for generalization of the RBM model to unseen items, our experiments suggest that the RBM choice model, without attributes, can learn the typical choice phenomena from a given choice set and generalize it to unseen choice sets.

## Acknowledgements

A part of this research is supported by JST, CREST.

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
