[Supplementary Material]

# Supplementary material for "Restricted Boltzmann machines modeling human choice"

**Takayuki Osogami**
IBM Research - Tokyo
osogami@jp.ibm.com

**Makoto Otsuka**
IBM Research - Tokyo
motsuka@ucla.edu

## Abstract

This document is the supplementary material for T. Osogami and M. Otsuka, "Restricted Boltzmann machines modeling human choice," appearing in *Advances in Neural Information Processing Systems 27 (NIPS 2014)*.

## A  Details about the RBM choice model

### A.1  Training algorithms for restricted Boltzmann machines

The parameters, $\theta \equiv (T, U, b)$, of the RBM choice model can be learned by a training algorithm in such a way that the log-likelihood of given training dataset, $\mathcal{D}$, is maximized. Here, the training dataset is a collection of observed pairs of a choice set and a selected item, $\mathcal{D} \equiv \{(\mathcal{X}_i, A_i)\}_i$. Existing training algorithms can be classified into discriminative ones, generative ones, or their hybrid, depending on what log-likelihood is maximized. For the RBM choice model, we find that the discriminative training algorithm runs faster and tends to learn the parameters more effectively than generative or hybrid training algorithms.

A discriminative training algorithm [102] updates $\theta$ in the direction of the gradient of the log-likelihood,

$$\sum_{(\mathcal{X},A)\in\mathcal{D}} \nabla_\theta \log p(A|\mathcal{X}). \tag{30}$$

To compute this gradient, let

$$g_\theta(X, \mathcal{X}) \equiv \frac{\sum_h P_\theta((v^{\mathcal{X}}, w^X), h)\, \nabla_\theta E_\theta((v^{\mathcal{X}}, w^X), h)}{\sum_{h'} P_\theta((v^{\mathcal{X}}, w^X), h')}.$$

We then have

$$\nabla_\theta \log p(A|\mathcal{X}) = -g_\theta(A, \mathcal{X}) + \frac{\sum_{X\in\mathcal{X}}\sum_h P_\theta((v^{\mathcal{X}}, w^X), h)\, g_\theta(X, \mathcal{X})}{\sum_{X\in\mathcal{X}}\sum_h P_\theta((v^{\mathcal{X}}, w^X), h)}. \tag{31}$$

The gradient (31) can be computed in the time that grows linearly with $|\mathcal{X}|$ and $|\mathcal{H}|$.

A generative training algorithm updates $\theta$ in the direction of the gradient of the log-likelihood,

$$\sum_{(\mathcal{X},A)\in\mathcal{D}} \nabla_\theta \log \sum_h P_\theta((v^{\mathcal{X}}, w^A), h). \tag{32}$$

Exact evaluation of this gradient is often intractable and requires some approximation scheme such as contrastive divergence [101]. A hybrid training algorithm considers a convex combination of the gradient in (32) and the gradient in (31). The best training algorithm appears to depend on particular problems [103].

In our experiments, we train the RBM choice model with the discriminative training algorithm with stochastic gradient descent using mini-batches. The training dataset, $\mathcal{D}$, is first divided into mini-batches of a given size. Then the parameters, $\theta$, are updated as

$$\theta \quad \leftarrow \quad \theta + \eta \sum_{X \in \mathcal{B}} \nabla_\theta \log p(X|\mathcal{X}) \tag{33}$$

for each mini-batch, $\mathcal{B}$, where $\eta$ is the learning rate. The training dataset can be used multiple times until the values of the parameters converge.

## A.2 Extensions of the RBM choice model

Our discussion in Section 2 motivates an extension of the RBM choice model. We now consider the bias, $b_k^{\mathrm{hid}}$, for a hidden unit, $k \in \mathcal{K}$. Also, for a visible unit, $X \in \mathcal{I}$, in the part representing the choice set, let $b_X^{\mathrm{set}}$ be the bias. For this full RBM choice model, the choice rate of $A$ from $\mathcal{X}$ is

$$\lambda'(A|\mathcal{X}) = \exp(b_A) \exp\left(b_{\mathcal{X}}^{\mathrm{set}}\right) \prod_{k \in \mathcal{K}} \left(1 + \exp((T_{\mathcal{X}}^k + U_A^k + b_k^{\mathrm{hid}}))\right), \tag{34}$$

where we define $b_{\mathcal{X}}^{\mathrm{set}} \equiv \sum_{X \in \mathcal{X}} b_X^{\mathrm{set}}$.

The factor of $\exp(b_{\mathcal{X}}^{\mathrm{set}})$ in (34) is canceled out when the choice rate is used in the choice probability (1). This factor can, however, become relevant when we want to model the choice rate itself for example to study the volume of sales per unit time. Namely, the choice rate can be used as a parameter of a stochastic process, such as a Poisson process, that generates a sequence of purchases. Then the choice rate can be interpreted as the expected volume of sales per unit time.

In (34), $b_k^{\mathrm{hid}}$ cannot be determined in the RBM choice model of selecting exactly one item. For each $k$, let

$$\tilde{U}_A^k \quad \equiv \quad U_A^k + b_k^{\mathrm{hid}}, \forall A \in \mathcal{I}. \tag{35}$$

We can thus replace the sum, $U_A^k + b_k^{\mathrm{hid}}$ with $\tilde{U}_A^k$ for each $A, k$, which is equivalent to setting $b_k^{\mathrm{hid}} = 0, \forall k$. The bias, $b_k^{\mathrm{hid}}$, can, however, become relevant when we consider selecting multiple items. In this case, $U_A^k$ in (34) becomes $\sum_{A \in \mathcal{A}} U_A^k$ for a set of selected items, $\mathcal{A}$, and then $b_k^{\mathrm{hid}}$ can play a role.

# B  Additional experimental results

## B.1 Details of the experimental results of Section 4

Figure 4 shows details of the results from the experiments in Section 4.

## B.2 Experimental results with artificial dataset

Consider the probability distribution shown in Figure 5 (a). Here, we have five items: $\mathcal{I} \equiv \{A, B, C, D, S\}$. The choice probabilities are designed to represent the typical choice phenomena for the representative choice sets shown in Figure 1. The similarity effect can be seen by comparing the choice probabilities for $\{A, B\}$ and those for $\{A, B, S\}$. Namely, $S$ is similar to $A$ and steals the market share only from $A$: $p(A|\{A, B\}) = 0.6$, $p(A|\{A, B, S\}) = 0.3$, and $p(B|\{A, B\}) = p(B|\{A, B, S\}) = 0.4$. Likewise, the attraction effect can be seen with $\{A, B\}$ and $\{A, B, D\}$. The compromise effect can be seen with $\{A, C\}$, $\{B, C\}$, and $\{A, B, C\}$.

We generate a dataset based on the probability distribution shown in Figure 5 (a). Specifically, for each of the six choice set, we generate 10 samples of selected items. Our dataset, $\mathcal{D}$, thus consists of 60 pairs of the choice set, $\mathcal{X}$, and the selected item, $X$. To reduce the variance in the results of experiments, the 10 samples are deterministically generated, so that the fraction of each selected item equals the corresponding probability in Figure 5 (a). For example, for $\mathcal{X} = \{A, B\}$, we select $X = A$ in six samples and $X = B$ in four samples.

Given $\mathcal{D}$, we train the RBM choice model by the use of discriminative training algorithm with stochastic gradient descent with a mini-batch of size 1 and learning rate of $\eta = 0.01$. Specifically,

Figure 4: Detailed view of Figure 3 (b). The left figure shows the average KL divergence for the training dataset, and the right figure shows the average KL divergence for the test dataset, where the average is over five iterations (with random initialization of parameters) for each test case (choice set). A red lines show the average KL divergence for the choice set with a car, and a blue line show that without a car. Although the legend in each figure shows only the ones with red or the ones with blue for readability, each figure shows the results for both with and without a car.

Figure 5: The choice probabilities given by the MLM (b) and the RBM-$n$ (d)-(f) that are trained based on the target distribution (a). Each bar represents the choice probabilities of the items from the choice set that is indicated below the bar.

the values of the parameters are updated according to (33) for each of the 60 pairs of $(\mathcal{X}, X)$ in $\mathcal{D}$ in the uniformly random order. This update with the 60 pairs is repeated 5,000 times to obtain the quality of the results to be presented. The initial values of the biases are set $b = 0$. The initial values of the elements of $T$ and $U$ are selected independently from the uniform distribution over $(-0.1, 0.1)$.

We vary $|\mathcal{K}|$, the number of hidden units, and examine how well we can recover the target distribution in Figure 5 (a) by training the parameters of the RBM choice model from the samples generated from the target distribution. Here, we refer to the RBM choice model with $|\mathcal{K}| = n$ as RBM-$n$. When $|\mathcal{K}| = 0$, the RBM choice model is reduced to the MLM. RBM-0 is thus called MLM.

The MLM is incapable of representing the typical choice phenomena, which can be seen in Figure 5 (b). For example, the target distribution exhibits the similarity effect (13). Specifically, we have $\psi_{A,B,S,\mathcal{X}}^{(\text{sim})} = 2$ for $\mathcal{X} \equiv \{A, B\}$, because $p(A|\mathcal{X}) = 0.6$, $p(A|\mathcal{X} \cup \{S\}) = 0.3$, and $p(B|\mathcal{X}) = p(B|\mathcal{X} \cup \{S\}) = 0.4$. However, the trained MLM has $\psi_{A,B,S,\mathcal{X}}^{(\text{sim})} = 1$ for $\mathcal{X} \equiv \{A, B\}$, because

$$\frac{p(A|\mathcal{X})}{p(B|\mathcal{X})} = \frac{p(A|\mathcal{X} \cup \{S\})}{p(B|\mathcal{X} \cup \{S\})} \approx 1.86, \tag{36}$$

where $p(A|\mathcal{X}) \approx 0.65$, $p(B|\mathcal{X}) \approx 0.35$, $p(A|\mathcal{X} \cup \{S\}) \approx 0.455$, and $p(B|\mathcal{X} \cup \{S\}) \approx 0.455$. Also, we can observe the attraction effect (20) in the target distribution, while the inequality in (20) is reversed in the trained MLM. Furthermore, (23) holds in the target distribution (i.e., the compromise effect), while the inequalities in (23) become equalities in the trained MLM.

A hidden unit greatly enhances the capability of the RBM choice model. Figure 5 (c) shows that the trained RBM-1 represents the typical choice phenomena. In the trained RBM-1, we can observe the similarity effect (13), the attraction effect (20), and the compromise effect (23). In fact, the trained RBM-1 quantitatively well approximates the target distribution. Only significant error can be seen in $p(\cdot|\{B, C\})$ and $p(\cdot|\{A, B, C\})$.

Taking a closer look, we can observe small error bars in Figure 5 (b)-(f). An error bar shows the sample standard deviation of the results from 10 runs, where the initial values of the parameters, $T, U$, are re-sampled independently in each run. The small error bars suggest the limitation of the RBM-1 model, rather than the training algorithm, in exactly matching the target distribution.

Figure 5 (d) shows that the trained RBM-2 better approximates the target distribution than the trained RBM-1. The error is now negligible for any choice set. This means that two hidden units suffice to represent all of the three typical choice phenomena. Recall that our theorems only suggest that one hidden unit is sufficient to represent each of the typical choice phenomena. In practice, each hidden unit contributes to representing multiple typical choice phenomena, and each typical choice phenomenon is represented by the superposition of the effects from multiple hidden units.

Adding further hidden units does not hurt the quality of the trained RBMs. The running time of the training algorithm is slightly increased with the additional hidden units. For example, MLM requires about 90 seconds for training, while RBM-4 requires about 120 seconds.

## C  Proofs

*Proof of Theorem 1.*  For $B \neq A$, we let $U_B^{\hat{k}} \to -\infty$ to obtain

$$\hat{\lambda}(B|\mathcal{Y}) = \lambda(B|\mathcal{Y}) \left(1 + \exp\left(T_{\mathcal{Y}}^{\hat{k}} + U_B^{\hat{k}}\right)\right) \tag{37}$$

$$\to \lambda(B|\mathcal{Y}) \tag{38}$$

for any $\mathcal{Y}$. This establishes the second part of (18).

To prove the first part of (18), let $T_X^{\hat{k}} = 0, \forall X \neq S$. Because $S \notin \mathcal{X}$, we have

$$\hat{\lambda}(A|\mathcal{X} \cup \{S\}) = \lambda(A|\mathcal{X} \cup \{S\}) \left(1 + \exp\left(T_S^{\hat{k}} + U_A^{\hat{k}}\right)\right) \tag{39}$$

$$\hat{\lambda}(A|\mathcal{X}) = \lambda(A|\mathcal{X}) \left(1 + \exp\left(U_A^{\hat{k}}\right)\right). \tag{40}$$

These two expressions give us

$$\frac{\hat{\lambda}(A|\mathcal{X} \cup \{S\})}{\hat{\lambda}(A|\mathcal{X})} = \frac{1 + \exp\left(T_S^{\hat{k}} + U_j^{\hat{k}}\right)}{1 + \exp\left(U_A^{\hat{k}}\right)} \frac{\lambda(A|\mathcal{X} \cup \{S\})}{\lambda(A|\mathcal{X})}. \tag{41}$$

Because the right-hand side of (41) is monotonically increasing with $T_S^{\hat{k}}$, it can take an arbitrary large value by letting $T_S^{\hat{k}} \to \infty$. Thus, we have

$$\lim_{T_S^{\hat{k}} \to \infty} \frac{\hat{\lambda}(A|\mathcal{X} \cup \{S\})}{\hat{\lambda}(A|\mathcal{X})} = \infty. \tag{42}$$

The corresponding lower limit is given by letting $T_S^{\hat{k}} \to -\infty$:

$$\lim_{T_S^{\hat{k}} \to -\infty} \frac{\hat{\lambda}(A|\mathcal{X} \cup \{S\})}{\hat{\lambda}(A|\mathcal{X})} = \frac{1}{1 + \exp(U_A^{\hat{k}})} \frac{\lambda(A|\mathcal{X} \cup \{S\})}{\lambda(A|\mathcal{X})}. \tag{43}$$

Because (41) is continuous with $T_S^{\hat{k}}$, the left-hand side of (41) can take any value between the lower limit (43) and the upper limit (42). The lower limit (43) can be made arbitrarily close to 0 by letting $U_A^{\hat{k}} \to \infty$. This establishes the first part of (18). $\qquad \square$

*Proof of Theorem 2.* As we have seen in the proof of Theorem 1, we can obtain (38) for any $\mathcal{Y}$ by letting $U_B^{\hat{k}} \to -\infty$ for $B \neq A$. This establishes the second part of (22).

To prove the first part of (22), let $T_X^{\hat{k}} = 0, \forall X \neq D$. Then we have

$$\hat{\lambda}(A|\mathcal{X}) = \lambda(A|\mathcal{X}) \left(1 + \exp\left(U_A^{\hat{k}}\right)\right) \tag{44}$$

$$\hat{\lambda}(A|\mathcal{X} \cup \{D\}) = \lambda(A|\mathcal{X} \cup \{D\}) \left(1 + \exp\left(T_D^{\hat{k}} + U_A^{\hat{k}}\right)\right). \tag{45}$$

Thus, by (17) and $D \notin \mathcal{X}$, we obtain

$$\hat{\psi}_{A,D,\mathcal{X}}^{(\text{att})} \equiv \frac{\hat{p}(A|\mathcal{X} \cup \{D\})}{\hat{p}(A|\mathcal{X})} \tag{46}$$

$$\to \frac{\dfrac{\lambda(A|\mathcal{X} \cup \{D\}) \left(1 + \exp\left(T_D^{\hat{k}} + U_A^{\hat{k}}\right)\right)}{\sum_{j \in \mathcal{X} \cup \{D\}} \lambda(j|\mathcal{X} \cup \{D\}) + \lambda(A|\mathcal{X} \cup \{D\}) \exp(T_D^{\hat{k}} + U_A^{\hat{k}})}}{\dfrac{\lambda(A|\mathcal{X}) \left(1 + \exp\left(U_A^{\hat{k}}\right)\right)}{\sum_{i \in \mathcal{X}} \lambda(i|\mathcal{X}) + \lambda(A|\mathcal{X}) \exp(U_A^{\hat{k}})}} \tag{47}$$

$$= \frac{\dfrac{\sum_{i \in \mathcal{X}} \lambda(i|\mathcal{X})}{\lambda(A|\mathcal{X})} + \exp(U_A^{\hat{k}})}{1 + \exp(U_A^{\hat{k}})} \frac{1 + \exp(T_D^{\hat{k}} + U_A^{\hat{k}})}{\dfrac{\sum_{j \in \mathcal{X} \cup \{D\}} \lambda(j|\mathcal{X} \cup \{D\})}{\lambda(A|\mathcal{X} \cup \{D\})} + \exp\left(T_D^{\hat{k}} + U_A^{\hat{k}}\right)} \tag{48}$$

$$= \frac{\dfrac{1}{p(A|\mathcal{X})} + \exp\left(U_A^{\hat{k}}\right)}{1 + \exp\left(U_A^{\hat{k}}\right)} \frac{1 + \exp\left(T_D^{\hat{k}} + U_A^{\hat{k}}\right)}{\dfrac{1}{p(A|\mathcal{X} \cup \{D\})} + \exp\left(T_D^{\hat{k}} + U_A^{\hat{k}}\right)} \tag{49}$$

in the limit of $U_B^{\hat{k}} \to -\infty, \forall B \neq A$.

Because $0 \le p(A|\mathcal{X} \cup \{D\}) \le 1$, the right-hand side of (49) is non-decreasing with $T_D^{\hat{k}}$ (this can be easily verified by taking the derivative with respect to $T_D^{\hat{k}}$). The lower limit of $\hat{\psi}_{A,D,\mathcal{X}}^{(\text{att})}$ is given by

$$\lim_{T_D^{\hat{k}} \to -\infty} \hat{\psi}_{A,D,\mathcal{X}}^{(\text{att})} = \frac{\frac{1}{p(A|\mathcal{X})} + \exp\left(U_A^{\hat{k}}\right)}{1 + \exp(U_A^{\hat{k}})} p(A|\mathcal{X} \cup \{D\}). \tag{50}$$

The corresponding upper limit is given by

$$\lim_{T_D^{\hat{k}} \to \infty} \hat{\psi}_{A,D,\mathcal{X}}^{(\text{att})} = \frac{\frac{1}{p(A|\mathcal{X})} + \exp\left(U_A^{\hat{k}}\right)}{1 + \exp(U_A^{\hat{k}})}. \tag{51}$$

Because $0 \leq p(A|\mathcal{X}) \leq 1$, the right-hand sides of (50) and (51) are non-increasing with $U_A^{\hat{k}}$. The lower limit of $\hat{\psi}_{A,D,\mathcal{X}}^{(\mathrm{att})}$ is thus given by

$$\lim_{U_A^{\hat{k}} \to \infty} \lim_{T_D^{\hat{k}} \to -\infty} \hat{\psi}_{A,D,\mathcal{X}}^{(\mathrm{att})} = p(A|\mathcal{X} \cup \{D\}). \tag{52}$$

The corresponding upper limit is given by

$$\lim_{U_A^{\hat{k}} \to -\infty} \lim_{T_D^{\hat{k}} \to \infty} \hat{\psi}_{A,D,\mathcal{X}}^{(\mathrm{att})} = \frac{1}{p(A|\mathcal{X})}. \tag{53}$$

These establishes the condition of the first part of (22) and completes the proof. $\square$

*Proof of Theorem 3.* As we have seen in the proof of Theorem 1, we can obtain (38) for any $\mathcal{Y}$ by letting $U_X^{\hat{k}} \to -\infty, \forall X \neq C$. This establishes the second part of (29).

To prove the first part of (29), let $T_X^{\hat{k}} = 0, \forall X \notin \{A, B\}$, $T_X^{\hat{k}} = 2M$ for $X \in \{A, B\}$, and $U_C^{\hat{k}} = -3M$, where $M$ is a constant that we will vary in the following. With these settings of $T$ and $U$, we have

$$\hat{\lambda}(C|\mathcal{X}) = \lambda(C|\mathcal{X})\,(1 + \exp(M)) \tag{54}$$

$$\hat{\lambda}(C|\mathcal{X} \setminus \{X\}) = \lambda(C|\mathcal{X} \setminus \{X\})\,(1 + \exp(-M)) \tag{55}$$

for $X = A, B$, because $A, B, C \in \mathcal{X}$.

Let $\hat{\phi}_{A,B,C,\mathcal{X}}$ be defined analogously to $\phi_{A,B,C,\mathcal{X}}$ but with $\hat{\lambda}$. Then we have

$$\hat{\phi}_{A,B,C,\mathcal{X}} = \frac{\dfrac{\hat{\lambda}(C|\mathcal{X})}{\sum_{X \in \{A,C\}} \hat{\lambda}(X|\mathcal{X})}}{\dfrac{\hat{\lambda}(C|\mathcal{X} \setminus \{B\})}{\sum_{X \in \{A,C\}} \hat{\lambda}(X|\mathcal{X} \setminus \{B\})}} \tag{56}$$

$$= \frac{\dfrac{\lambda(C|\mathcal{X})(1 + \exp(M))}{\sum_{X \in \{A,C\}} \lambda(X|\mathcal{X}) + \lambda(C|\mathcal{X}) \exp(M)}}{\dfrac{\lambda(C|\mathcal{X} \setminus \{B\})(1 + \exp(-M))}{\sum_{X \in \{A,C\}} \lambda(X|\mathcal{X} \setminus \{B\}) + \lambda(C|\mathcal{X} \setminus \{B\}) \exp(-M)}} \tag{57}$$

$$= \frac{1 + \exp(M)}{1 + \exp(-M)} \frac{\dfrac{1}{q_{AC}(C|\mathcal{X} \setminus \{B\})} + \exp(-M)}{\dfrac{1}{q_{AC}(C|\mathcal{X})} + \exp(M)} \tag{58}$$

$$= \frac{q_{AC}(C|\mathcal{X})}{q_{AC}(C|\mathcal{X} \setminus \{B\})} \frac{\exp(M) + q_{AC}(C|\mathcal{X} \setminus \{B\})}{1 + q_{AC}(C|\mathcal{X}) \exp(M)} \tag{59}$$

Taking the derivative with respect to $M$, we find that

$$\frac{\partial \hat{\phi}_{A,B,C,\mathcal{X}}}{\partial M} = \frac{q_{AC}(C|\mathcal{X}) \exp(M)}{q_{AC}(C|\mathcal{X} \setminus \{B\})} \frac{1 - q_{AC}(C|\mathcal{X} \setminus \{B\})\, q_{AC}(C|\mathcal{X})}{(1 + q_{AC}(C|\mathcal{X}) \exp(M))^2} \tag{60}$$

$$\geq 0. \tag{61}$$

Hence, $\hat{\phi}_{A,B,C,\mathcal{X}}$ is non-decreasing with $M$. The lower limit of $\hat{\phi}_{A,B,C,\mathcal{X}}$ is given by

$$\lim_{M \to -\infty} \hat{\phi}_{A,B,C,\mathcal{X}} = q_{AC}(C|\mathcal{X}). \tag{62}$$

The corresponding upper limit is given by

$$\lim_{M \to \infty} \hat{\phi}_{A,B,C,\mathcal{X}} = \frac{1}{q_{AC}(C|\mathcal{X} \setminus \{B\})}. \tag{63}$$

Because $\hat{\phi}_{A,B,C,\mathcal{X}}$ is continuous with $M$, $\hat{\phi}_{A,B,C,\mathcal{X}}$ can take an arbitrary value in

$$\left( q_{AC}(C|\mathcal{X}), \frac{1}{q_{AC}(C|\mathcal{X} \setminus \{B\})} \right). \tag{64}$$

Exchanging the role of $A$ and $B$ in (59), we can see that

$$\hat{\phi}_{B,A,C,\mathcal{X}} = \frac{q_{BC}(C|\mathcal{X})}{q_{BC}(C|\mathcal{X} \setminus \{A\})} \frac{\exp(M) + q_{BC}(C|\mathcal{X} \setminus \{A\})}{1 + q_{BC}(C|\mathcal{X}) \exp(M)} \tag{65}$$

is non-decreasing with $M$ and can take arbitrary value in

$$\left( q_{BC}(C|\mathcal{X}), \frac{1}{q_{BC}(C|\mathcal{X} \setminus \{A\})} \right). \tag{66}$$

Because both of $\hat{\phi}_{A,B,C,\mathcal{X}}$ and $\hat{\phi}_{B,A,C,\mathcal{X}}$ are non-decreasing with $M$, the minimum of these quantities (i.e., $\hat{\psi}_{A,B,C,\mathcal{X}}$) is also non-decreasing with $M$ and, by (64) and (66), can take an arbitrary value in $(\underline{q}, 1/\overline{q})$. This establishes the theorem. $\qquad\square$