[Reviews · NeurIPS 2014]

Submitted by Assigned_Reviewer_6

In this paper the authors propose a flexible RBM choice model that can be used to learn the typical choice phenomena, including the similarity effect, the attraction effect, and the compromise effect. The author also show that their choice model is equivalent to a restricted Boltzmann machine whose parameters can be learned efficiently.

Quality:
The paper is technically sound. It would be nice if the author could discuss more limitations of this work. My main concern is the limited novelty of this paper.

Clarity:
Overall, this paper is clearly. However, it might be better if the authors could explain each term clearly when it is mentioned for the first time. For example, the weight parameters in equation (3) could be explained immediately after the equation. In addition, the authors may consider removing figure 1 since it does not help me understanding the typical choice phenomena. And it is more logical to discuss related work in the introduction section.

Originality

I think the paper explored a good approach solving a open problem. However, I don’t see how this paper explored into new directions of research in the NIPS community.

Significance

The paper presents an important topic and suggests a simple approach that is computationally feasible and understandable.

Summary: This is an interesting paper. I recommend this paper for acceptance if the human choice modeling problem is an important open problem to the NIPS community.

Submitted by Assigned_Reviewer_22

This paper presents how restricted Boltzmann machines can be used to learn human choice preferences. Because restricted Boltzmann machines do not force choices to be independent of the set of alternatives, the RBM model can represent typical human choice phenomena that cannot be modeled by the multinomial logistic models that are used as a standard model of human choice. The work proves that the RBM choice model can model several standard features of human choices, and that RBMs can flexibly model a range of strengths of these features. The evaluation includes synthetic data (in the supplementary material) as well as a comparison of RBM versus MLM models for a dataset on real human choices.

Overall, the model seems technically sound and is indeed able to model typical human choice phenomena. One concern I had was whether the model is too flexible: are there features of human choices that this model would not be able to model, or parameter values that we might expect to share across datasets to constrain the learned parameters? The presentation does not seem to suggest that RBMs can be used as a cognitive model, but simply that they can predict human choices; this may make the concern about falsifiability less pressing, but it seems like some discussion of whether flexibility could be a negative feature of the model would be helpful.The theorems that are proven do demonstrate that given only a single hidden unit, the choice phenomena can be represented up to a particular range of strengths, but the evaluation is not restricted to using a single hidden unit and performance on test with a single hidden unit does not appear to be much better than not using a hidden unit. Since the theorems only show that a single typical choice phenomenon can be represented using one hidden unit, it makes sense that more hidden units may be required in the simulations (thank you for this clarification in the rebuttal). However, it seems that the discussion should then connect the theoretical and empirical work to make sense of what we can learn about human choice and how much we can generalize to new choices given the extreme flexibility of the actual models used in the simulations.

One of my concerns related to the flexibility of the model was whether the RBM choice model could be used to predict choices about new items based on their features. This is addressed in the discussion with the sentence "The focus of this paper is in designing the fundamental structure of the RBM choice model and analyzing its fundamental properties, and the study about the RBM choice model with attributes is reported elsewhere." Thank you to the authors for clarifying that this is work in progress/not yet published - I think this sentence could be re-worded to make it more clear, but my concern about this paper standing on its own given the existence of the other work has been addressed. I do wonder, though, if it really makes sense to separate the two pieces of research.

In the evaluation, I would have appreciated more qualitative explanation of the meaning of the hidden units in the RBM (i.e., why here 4 hidden units is approximately equivalent in performance to 16); such explanation might be helpful in understanding whether we can draw any conclusions about human choices based on the learned RBM model or if the model is only descriptive. To me, it seems like learning something about human choices (rather than just describing them computationally) needs to be a key outcome of this paper, especially given the flexibility of RBMs that the authors note.

The qualitative clarification about generalizability in the rebuttal is helpful. I still believe it would strengthen the paper substantially to do more detailed testing than leave one out. In particular, it would strengthen the submission to more closely examine the effects of training set size and characteristics on test performance.

Overall, my main concern about this paper is on its focus: computationally describing human choice phenomena and predicting a new choice from existing choices without looking at what we might learn about choices from this model or how much overfitting is a problem. Learning about choices from the model seems likely to be most interesting to computational cognitive scientists, but the current paper doesn't address this issue; I think it could be addressed using the current work, and I would appreciate seeing discussion of it. The overfitting concern seems to be most relevant for the broader machine learning community, which might be interested mainly in the applications. I think the paper could be substantially strengthened by addressing these issues or by explaining why they are not in scope; this concern is why I think presenting this work with the attribute work might result in more impact.

Minor notes:
line 18 (and later): "the choice made by humans" is used several times in the paper - I think this should read "the choices made by humans", but maybe this is a specialized use?
line 375: Maglve20 -> Maglev20
line 377: such attraction effect -> such attraction effects
line 431: from given choice set -> from a given choice set
Summary: The paper presents interesting work about modeling human choice phenomena using RBMs and is technically sound. The impact of the research may be limited due to the highly flexible nature of the RBM model and lack of discussion on whether we can learn something about human choice based on this model.

Submitted by Assigned_Reviewer_45

This paper proposed a novel technique to model human choice behavior. This technique is based on Decision Field Theory and is the first model to be able to model Similarity effect, Attraction effect, and Compromise effect at the same time.

This work is significant because this technique is the first to fully model various aspects of Decision Field Theory. This technique is expected to be widely used because of its strength and simplicity. This work is based on Restricted Boltzmann Machine that is a well established generative model.

The paper is very clear and well written.

I think the only major weakness of the paper is the limited experiments.
Summary: + Well written and clear math
+ Significance of the problem
+ Simplicity of the technique
+ Scalability and potential to apply to large datasets
- limited experimentation

The experiments in the paper are performed on a dataset that is made up of only 45 probabilities (only 27 degrees of freedom). 25 or 26 values are used for training, while 1 or 2 scalar values are expected to be extracted from model. This is indeed very small for an RBM-based technique.

The authors could either use a larger dataset with more choices or use the same dataset with slightly different setup. Instead of feeding in 17 out of 18 questions and predicting one question, they could feed in 6 questions and predict 12 questions.

I think given the limitation of the experiments, the paper is still a good paper to be published.
Author Feedback
Author rebuttal: *** Responses to Assigned_Reviewer_22 ***

Re: Fexibility

The RBM choice model might indeed be too flexible as a model of human choice. This over-flexibility can potentially cause over-fit to training data, but such over-fit can be avoided by the use of drop out and other heuristics that have been proposed for general RBMs.

Re: Single hidden unit

Our theorem states that a single hidden node suffices to represent one of the typical choice phenomena, to the degree specified in the theorem, among a particular set of items. The RBM choice model might need more than one hidden node to represent more than one typical choice phenomenon and when more than two sets of items involve typical choice phenomena. Our theorems thus do not contradict with experimental results.

Re: RBM choice model with attributes

The RBM choice model with attributes **will be** reported elsewhere. This extension indeed depends significantly on this submission and will be published after this submission is published.

Re: Evaluation of the meaning of hidden units

Evaluation of hidden units of trained RBM choice model is indeed an interesting direction of future work.

Re: Figure 3

Figure 3 (b) shows the average and standard deviation of the results for 18 test cases. Namely, we first train the RBM choice model using all training data from 17 choice sets and then predict the choices on the 18th set, and this is repeated by varying the test cases in 18 different ways. On the other hand, Figure 3 (c) and Figure 3 (d) show the result for one of the 18 test cases (specifically, the test case consists of Train30 and Maglev20).
The generalization to unseen choice sets is possible when there are some clues. For example, the choice probabilities for a choice set of Train30 and Maglev20 can be inferred from the choice probabilities for a choice set of Train30, Maglev20, and Car as well as the impact of Car on the choice probabilities with other choice sets. Without such clues, the RBM choice model is unlikely to generalize to unseen choice sets.

*** Responses to Assigned_Reviewer_45 ***

We have no comments. Thank you very much for your careful review.

*** Responses to Assigned_Reviewer_6 ***

Re: Limitation

The RBM choice model might be too flexible as a model of human choice. This over-flexibility can potentially cause over-fit to training data, but such over-fit can be avoided by the use of drop out and other heuristics that have been proposed for general RBMs.

Re: New directions of research

The RBM choice model motivates an extension to choice models with deep Boltzmann machines or deep belief networks. Deep learning with these extended models have recently been shown to outperform other machine learning algorithms in those tasks where it is hard for machines to achieve the performance of humans. In practice, we choose or buy products not only based on simple attributes such as price and quantity but also based on complex features such as how they look (e.g., pictures in a menu) and how they read (e.g., narrative descriptions of products). It is an interesting direction to investigate if deep learning allows us to take into account these complex features in choice models.